# Tissue-Specific Expression of the Terpene Synthase Family Genes in *Rosa chinensis* and Effect of Abiotic Stress Conditions

**DOI:** 10.3390/genes13030547

**Published:** 2022-03-20

**Authors:** Yuhang Yan, Mouliang Li, Xiaoni Zhang, Weilong Kong, Mohammed Bendahmane, Manzhu Bao, Xiaopeng Fu

**Affiliations:** 1Key Laboratory of Horticultural Plant Biology, Ministry of Education, College of Horticulture and Forestry Sciences, Huazhong Agricultural University, Wuhan 430070, China; yuhang_Yan@webmail.hzau.edu.cn (Y.Y.); mouliang_Li@webmail.hzau.edu.cn (M.L.); mzbao@mail.hzau.edu.cn (M.B.); 2Guangdong Laboratory for Lingnan Modern Agriculture, Genome Analysis Laboratory of the Ministry of Agriculture, Agricultural Genomics Institute at Shenzhen, Chinese Academy of Agricultural Sciences, Shenzhen 518120, China; zhangxiaoni@webmail.hzau.edu.cn (X.Z.); weilong.Kong@whu.edu.cn (W.K.); 3Laboratoire Reproduction et Development des Plantes, Ecole Normale Supérieure Lyon, 520074 Lyon, France; mohammed.bendahmane@ens-lyon.fr

**Keywords:** rose, terpene synthase, genome-wide analysis, expression pattern, abiotic stress

## Abstract

Rose (*Rosa chinensis*) is one of the most famous ornamental plants worldwide, with a variety of colors and fragrances. Terpene synthases (TPSs) play critical roles in the biosynthesis of terpenes. In this work, we report a comprehensive study on the genome-wide identification and characterization of the TPS family in *R. chinensis*. We identified 49 *TPS* genes in the *R. chinensis* genome, and they were grouped into five subfamilies (TPS-a, TPS-b, TPS-c, TPS-g and TPS-e/f). Phylogenetics, gene structure and conserved motif analyses indicated that the *RcTPS* genes possessed relatively conserved gene structures and the RcTPS proteins contained relatively conserved motifs. Multiple putative *cis*-acting elements involved in the stress response were identified in the promoter region of *RcTPS* genes, suggesting that some could be regulated by stress. The expression profile of *RcTPS* genes showed that they were predominantly expressed in the petals of open flowers, pistils, leaves and roots. Under osmotic and heat stresses, the expression of most *RcTPS* genes was upregulated. These data provide a useful foundation for deciphering the functional roles of *RcTPS* genes during plant growth as well as addressing the link between terpene biosynthesis and abiotic stress responses in roses.

## 1. Introduction

Terpene synthases (TPSs) are important enzymes involved in the biosynthesis of terpenes, a class of compounds related to plant growth and development that attract insect pollinators and respond to multiple biotic or abiotic stresses [1,2,3,4]. Generally, the precursors of terpenoids are two isomeric C5 compounds, isopentenyl diphosphate (IPP) and dimethylallyl diphosphate (DMAPP) [5]. In most plants, IPP and DMAPP are the products of two separate pathways, the cytosolic mevalonic acid (MVA) pathway and the plastid methylerythritol phosphate (MEP) pathway [6]. IPP and DMAPP are subsequently condensed into farnesyl diphosphate (FPP), geranyl diphosphate (GPP), and geranylgeranyl diphosphate (GGPP), which are the direct precursors of terpenes (Figure 1). Finally, FPP is converted into sesquiterpenes (C15) in the cytosol, while GPP and GGPP are converted into monoterpenes (C10) and diterpenes (C20), respectively, in plastids by TPSs [3,7,8]. These terpene skeletons are further modified by various enzymes, such as cytochrome P450-dependent monooxygenases (CYPs), methyltransferases, dehydrogenases and acyltransferases, to form additional compounds [9,10].

TPS proteins are classified into two types, I and II, based on their structure and reaction mechanism [7,11,12]. Type I TPS proteins contain conserved DDXXD and NSE/DTE motifs near the C-terminus and are known to play important roles in the metal-dependent ionization of prenyl diphosphate substrates [13,14,15]. Type II TPS proteins contain a DXDD motif near the N-terminus that catalyzes the formation of copalyl diphosphate (CPP) through protonation-induced cyclization of GGPP [3,16]. The TPS family is phylogenetically divided into seven subfamilies, one of which, TPS-c, represents the type II TPSs, while the remaining subfamilies, TPS-a, TPS-b, TPS-d (gymnosperm-specific), TPS-e/f, TPS-g and TPS-h (specific to *Selaginella moellendorffii*), are type I TPS proteins [3,7,17].

To date, the TPS family has been identified in various plant species, such as *Arabidopsis thaliana* [18], *Vitis vinifera* [19], *Solanum lycopersicum* [20,21], *Malus domestica* [22], and *Eucalyptus grandis* [23], *Citrus sinensis* [24], *Setaria italica* [25], *Camellia sinensis* [26], *Dendrobium officinale* [27] and *Aquilaria sinensis* [28]. The number of *TPS* genes varies widely between different species. *V. vinifera* has the largest known *TPS* gene family with as many as 69 putatively functional *TPS* genes, while there is only one functional *TPS* gene that has been identified in *Physcomitrella patens* [29]. Previous studies have characterized several monoterpene synthases and predicted the TPS family genes in the rose [30,31]. However, to date, no comprehensive study of the TPS family has been reported in the rose.

*Rosa* spp., an economically important genus belonging to the *Rosaceae* family, are cultivated worldwide for their high ornamental properties and cosmetic and medicinal properties [32]. Modern roses have inherited many traits, mainly floral, including their scent, from both European and Chinese lineages through years of artificial hybridization [33]. Terpenoids, benzenoids/phenylpropanoids and fatty acid derivatives are the three major classes of scented compounds in *R. chinensis* [31]. In particular, monoterpenes represent up to 70% percent of the scent content in some cultivars [34]. Like other plant species, in roses the sesquiterpenes and norterpenes are synthesized in cytosol and plastids, respectively [30,35]. However, instead of being synthesized by TPSs in the plastids, the acyclic monoterpenes in roses are synthesized in the cytosol by a noncanonical enzyme named NUDX1, which is different from other plants [34] (Figure 1).

Roses have to cope with many abiotic stresses, such as heat stress and water deficit caused drought stress, which severely limit their growth and flowering [36,37]. Drought stress reduces the number of petals and the length of the flower bud of roses [38]. High-temperature conditions were reported to promote flower abscission, and to decrease anthocyanin pigmentation and the size of petals in the rose [39,40]. Volatile terpenes as constituents of floral scent not only play significant roles in attracting pollinator, but they have also been shown to be involved in the response to abiotic stresses in plants [1,26], which is mainly because some of the terpenes (such as isoprene and some monoterpenes) can maintain the stability of cell membrane structure under stress conditions and eliminate the damage caused by the stress-induced accumulation of reactive oxygen species (ROS) [41,42,43]. Thus, understanding how *TPS* genes respond to such stresses in roses will help to develop strategies to improve their tolerance.

In this report, we identified and analyzed TPS family members in *R. chinensis**,* a major modern roses progenitor, at the genome-wide level and investigated the gene structure, conserved motifs, phylogenetic relationships, tissue-specific expression profiles and expression patterns under abiotic stresses. The present research provides a valuable foundation that will help to understand the role of rose *TPS* genes during development and the stress response, as well their regulation and evolution, and will help to provide ideas for breeding new rose varieties with stronger resistance.

## 2. Materials and Methods

### 2.1. Plant Materials and Treatments

*R. chinensis* ‘Old Blush’ plants were grown in temperature and light-controlled growth chamber (25 °C, 60% relative humidity, 150 µE light intensity, 16 h light/8 h darkness). The plants were cultivated in separate pots (10 × 10 × 9 cm) containing a mixture of peat moss, vermiculite and perlite (3: 2: 1) and were irrigated twice a week. For heat treatment, 3-year-old rose plants were watered once before treatment and then incubated at 35 °C for 24 h, Petals were harvested at 0, 6, 12, 24 h post-treatment. For drought treatment, the plants were irrigated once before treatment using a 200 mM mannitol solution until it dripped out from the pot bottom. Petals were harvested at 0, 12, 24, 48 h post-treatment. All samples were immediately frozen in liquid nitrogen and stored at −80 °C until use. Total RNA was extracted using EASYspin plant RNA extraction kit (AidLab, Beijing, China) and genomic DNA was removed by on-column DNAse digestion. RNA was quantified using the NanoDrop 2000c spectrophotometer (Thermo Scientific, Wilmington, DE, USA) and RNA integrity was assessed by standard denaturing 1% TBE agarose gel electrophoresis. A total amount of 1 µg RNA was then reverse transcribed into cDNA using the TRUEscript RT MasterMix (AidLab, Beijing, China) with the following conditions: 42 °C for 20 min and 85 °C for 5 s. The cDNA was then diluted 10-fold and stored at −20 °C for qRT-PCR.

### 2.2. Identification of TPS Genes in R. chinensis

The reference rose genome sequence used in this study was downloaded from the *R. chinensis* ‘Old Blush’ genome website (https://lipmbrowsers.toulouse.inra.fr//pub/RchiOBHm-V2/, accessed on 24 February 2022) [31]. *A. thaliana* TPS (AtTPS) protein sequences were collected from the Arabidopsis Information Resource (https://www.arabidopsis.org/, accessed on 24 February 2022) [44] (Appendix A). AtTPS were used as query sequences by BLASTP searches [45] (e-value 1 × 10^−6^) against the Rose’s protein sequences to predict the TPS family genes in *R. chinensis*. In parallel, we downloaded two specific TPS domains (PF03936 and PF01397) in the Pfam database (http://pfam.xfam.org/, accessed on 24 February 2022) and used HMM search tool [46] to identify the rose *RcTPS* genes as well. The genes obtained by the two approaches above, were compared and each predicted gene was subsequently verified through the SMART (http://smart.embl.de/, accessed on 24 February 2022) and NCBI-CDD searches (https://www.ncbi.nlm.nih.gov/cdd/, accessed on 24 February 2022) to further confirm the completeness of the TPS domain. Subcellular localization was predicted using the tools Plant-mPLoc (http://www.csbio.sjtu.edu.cn/bioinf/plantmulti/, accessed on 24 February 2022) and pLoc-mPlant (http://www.jcibioinfo.cn/pLoc-mPlant/, accessed on 24 February 2022).

### 2.3. Chromosomal Distribution, Duplication Event of RcTPS Genes 

Duplication event of TPS family genes in rose genome was analyzed using MCScanX program with default parameters [47]. The chromosome position of each *RcTPS* gene was mapped on the assembled *R. chinensis* ‘Old Blush’ genome (https://lipmbrowsers.toulouse.inra.fr//pub/RchiOBHm-V2/, accessed on 24 February 2022) [31], using TBtools software [48].

### 2.4. Gene Structure, Conserved Motif and Phylogenetic Analysis

Multiple sequence alignments between the TPS protein sequences of *R. chinensis*, *A. thaliana*, *S. lycopersicum* and *M. domestica* were performed using MEGA7.0 software [49]. Neighbor-joining (NJ) method was used to construct the phylogenetic tree with bootstrap replications set to 1000 and other parameters set as default values. The information of exon-intron structures was downloaded from the *R. chinensis* ‘Old Blush’ genome website (https://lipmbrowsers.toulouse.inra.fr//pub/RchiOBHm-V2/, accessed on 24 February 2022) [31]. Conserved motifs of TPS proteins were analyzed using MEME program (http://www.meme-suite.org, accessed on 24 February 2022). The maximum number of predicted motifs in the parameters was set to 10, and the other parameters were set as default. The map of gene structure and conserved motifs was drawn using TBtools software.

### 2.5. Analysis of Cis-Acting Elements in the Promoter Region of RcTPS Genes

For each identified *RcTPS* gene, 2000-bp sequence located upstream ATG translation initiation codon, was extracted from the *R. chinensis* ‘Old Blush’ genome sequence [31], and was then submitted to the PlantCARE database (http://bioinformatics.psb.ugent.be/webtools/plantcare/html/, accessed on 24 February 2022) to predict *cis*-acting elements.

### 2.6. Gene Expression Analysis Based on Transcriptome Data

Raw transcriptome data from nine different tissues (roots, stems, leaves, stamens, pistils, green petals in the flower buds (FB_GP), color-changing petals in the flower buds (FB_CP) and pink petals in the flower buds (FB_PP), and pink petals of the open flowers (OF_PP)), were downloaded from NCBI under BioProject PRJNA546486 [50] and BioProject PRJNA351281 [51]. The SRA files were converted to FASTQ files using the SRA toolkit (http://www.ncbi.nlm.nih.gov/Traces/sra/, accessed on 24 February 2022). Then the FastQC software [52] was used to assess read quality, and Trimmomatic [53] was used to discard low quality portions of reads (QUALITY: 15, LEADING: 20, TRAILING: 20, MINLEN: 50, SLIDINGWINDOW: 5: 20). The reads were subsequently mapped to *R. chinensis* ‘Old Blush’ genome with HISTA2 [54] and the bam files were processed to analyze gene expression quantitatively using StringTie [55] with default parameters. Fragments per kilobase per million mapped reads (FPKM) values were used to evaluate transcript abundance of *RcTPS* genes. The heat map of expression profiles of *RcTPS* genes in different tissues was drawn using the TBtools software. The expression levels of *RcTPS* genes are expressed by log2-based FPKM with row standardization, other parameters were set as default parameters.

### 2.7. qRT-PCR Analysis of RcTPS Genes under Different Treatment

Primers used for quantitative real-time PCR (qRT-PCR) analysis were designed by NCBI primer blast (https://www.ncbi.nlm.nih.gov/tools/primer-blast/, accessed on 24 February 2022) according to the CDSs of genes (Appendix A). Reaction was performed using PowerUp SYBR Green Master Mix (ABI, Carlsbad, CA, USA). qRT-PCR was carried out in 384-well plates in a 10 μl volume on QuantStudio 6 and 7 Flex Real-Time PCR system (ABI, Carlsbad, CA, USA) with the following cycling parameters: heating at 95 °C for 2 min, 40 cycles of denaturation at 95 °C for 15 s, annealing at 60 °C for 15 s, and extension at 72 °C for 1 min. Each sample was conducted with three biological replicates and melt curves were generated to check the non-specific amplification for each well. The glyceraldehyde-3-phosphate dehydrogenase (*RcGAPDH*) was used as housekeeping gene to normalize samples [56]. The relative expression values and error values were calculated using the comparative CT(2^−ΔΔCT^) method [57]. Gene expression values were log2 transformed and the TBtools software was used to generate the heatmaps. Statistical analysis was performed using one-way ANOVA along with Duncan’s multiple range test using SAS version 9.4 (SAS Institute), and P <0.05 were considered significant.

## 3. Results

### 3.1. Identification, Classification and Properties of RcTPS Genes

To identify *TPS* genes in the *R. chinensis* genome, we performed a BLASTP search using known *A. thaliana* TPS protein sequences as queries. To further verify the identity of the detected rose TPS sequences, the hidden Markov model (HMM) profile of the conserved C-terminal domain (PF03936) and N-terminal (PF01397) domain of TPSs was downloaded from the Pfam database [58] and used as a query in the HMM search. All putative *TPS* genes were further confirmed by SMART [59] and NCBI-CDD searches for the presence of the complete domain. We identified 49 *TPS* gene candidates in the *R. chinensis* ‘Old Blush’ genome (named *RcTPS01* to *RcTPS49*; Table 1). The lengths of the proteins encoded by these *RcTPS* genes varied from 455 (RcTPS07) to 857 (RcTPS49) amino acids. The predicted isoelectric points ranged from 5.06 (RcTPS43) to 8.91 (RcTPS47). The protein molecular weights varied between 52.89 kDa (RcTPS07) to 98.52 kDa (RcTPS49). According to two widely used subcellular localization predictors (Plant-mPLoc [60] and pLoc-mPlant [61]), 14 RcTPS proteins were predicted to be chloroplastic TPSs. The remaining 35 RcTPS proteins were predicted as chloroplastic or cytoplasmic TPSs.

### 3.2. Chromosomal Distribution, Duplication Event of RcTPS Genes

Genome chromosomal location analysis indicated that the 49 *RcTPS* genes were distributed on the seven rose chromosomes (Figure 2). Chromosome 5 contained the highest number (20) of *RcTPS* genes. So far only one *TPS* gene was located on chromosome 4, and two *TPS* genes were located on chromosome 2, the second longest rose chromosome, suggesting that there is no apparent correlation between chromosome size and *RcTPS* gene distribution.

To further study the evolution of rose *RcTPS* genes, we analyzed tandem duplication events and segmental duplication events on all seven chromosomes using the MCScanX tool. Four tandem duplication events with seven *RcTPS* genes were identified in the *R. chinensis* genome, whereas no pairs were confirmed as segmental duplications, indicating that tandem duplication events mainly contribute to the expansion of the TPS family in roses.

### 3.3. Phylogenetic Analysis of RcTPS Genes in R. chinensis

To investigate the classification and evolutionary relationships between RcTPS proteins, an unrooted phylogenetic tree was constructed based on alignments of the TPS proteins of *R. chinensis*, *A. thaliana*, *S. lycopersicum* and *M. domestica* using the neighbor-joining (NJ) method in MEGA 7.0 (Figure 3). The 49 *RcTPS* genes were classified into five subfamilies (TPS-a, TPS-b, TPS-c, TPS-g, TPS-e/f) based on phylogenetic analysis. Among the 49 RcTPS proteins, 48 members (RcTPS01 to RcTPS48) with high similarity could be classified as type I TPSs, whereas the remaining RcTPS49, which clustered in the same branch as AtTPS31, was classified as a type II TPS. Six TPS members (RcTPS09, RcTPS13, RcTPS19, RcTPS20, RcTPS33, RcTPS46) formed the TPS-b subfamily, 4 members (RcTPS35, RcTPS37, RcTPS40, RcTPS41) formed the TPS-g subfamily, 4 members (RcTPS02, RcTPS29, RcTPS31, RcTPS47) formed the TPS-e/f subfamily, and the remaining 34 members formed the TPS-a subfamily. The classification results were consistent with the previous report in Arabidopsis [18].

### 3.4. Gene Structure and Conserved Motif Analysis

To better understand the gene structure and evolutionary relationships, the intron–exon structure of each *RcTPS* gene was analyzed (Figure 4). The number of exons ranged from 5 to 14, and most of the *RcTPS* genes that clustered in the same subfamily showed a similar gene structure. *RcTPS31* and *RcTPS49* had the highest number of exons (14 exons), whereas *RcTPS40* contained the fewest exons (5 exons). Except for *RcTPS07*, *RcTPS27*, *RcTPS35*, *RcTPS40* and *RcTPS41*, all 44 remaining *RcTPS* genes of the subfamilies TPS-a, TPS-b and TPS-g contained 7 coding exons. Moreover, the *RcTPS* genes in the TPS-e/f subfamily contained more exons (10–14), which is similar to previously reported data in *V. vinifera*, *D. officinale* and *E. grandis* [19,23,27].

To further clarify the functional characteristics and specificities among the RcTPS proteins, 10 conserved protein motifs were identified using the MEME motif search tool. Each identified motif contained between 21 and 50 amino acids (Appendix A). Most RcTPS proteins in the same subfamily shared similar motif structures (Figure 5). Motif-1, motif-4, motif-6 and motif-8 were found in all RcTPS proteins. Motif-1 harbors an aspartate-rich region, DDXXD, representing the most typical conserved motif contained in the C-terminal of TPS proteins. Motif-7 was another important motif of the C-terminal domain in TPS proteins, which could be found in almost all RcTPS proteins of the subfamilies TPS-a, TPS-b, TPS-g and TPS-e/f. Previously, this motif, characterized by the consensus sequence (L,V)(V,L,A)(N,D)D(L,I,V)X(S,T)XXXE, was named NSE/DTE [62]. Both the DDXXD and NSE/DTE motifs are known to play an important role in cleaving prenyl diphosphate substrates by binding Mg^2+^ or Mn^2+^ at the C-terminus [1,14,15]. In rose TPSs, the DDXXD motif shows a higher level of conservation than the NSE/DTE motif. In addition, the arginine-tryptophan motif RRX_8_W (motif-9) was found in the N-terminus of all RcTPS-a and RcTPS-b proteins. This motif is known to play an essential role in the catalysis of monoterpene cyclization [3,15].

### 3.5. Analysis of Cis-Acting Elements in the Promoter Region of RcTPS Genes

Most of the functional *cis*-acting elements is concentrated within proximal promoters, usually in the region from − 1,000-bp to + 200-bp relative to the transcription start site (TSS) [63]. In this study, we analyzed the 2000-bp nucleotide sequence located upstream of the ATG initiation codon for all *RcTPS* genes using the PlantCARE database [64]. We identified three categories of *cis*-acting regulatory elements that share similarities with previously reported motifs associated with plant growth and development, phytohormone response and stress responses (Figure 6). *Cis*-acting regulatory elements involved in plant growth and development mainly include flowering regulation elements (CCAAT-box, AT-rich element, and MRE), endosperm-specific expression elements (GCN4-motif and AAGAA-motif), a shoot and root meristem expression element (CAT-box), a seed-specific regulation element (RY-element), and a shoot-specific expression element (As-1 element). Phytohormone-responsive elements included a salicylic acid-responsive element (TCA-element), gibberellin-responsive elements (P-boxes, GARE motifs, and TATC boxes), an auxin-responsive element (TGA-element), an ethylene-responsive element (ERE), an abscisic acid-responsive element (ABRE) and jasmonic acid-responsive elements (TGACG motifs and MYCs). The stress response category included defense and stress responsiveness (TC-rich repeats), wounding responses (WRE3 and WUN-motif), anaerobic induction (ARE), drought response (MBS), dehydration response (DRE), low temperature response (LTR) and stress response (STRE) (Figure 6). These data indicate that *RcTPS* genes are likely regulated by multiple phytohormones and abiotic stresses.

### 3.6. Expression Patterns of RcTPS Genes in Different Tissues of R. chinensis

The transcriptome data of nine rose tissues, including roots, stems, leaves, stamens, pistils and flowers, at different developmental stages were downloaded from the NCBI to analyze the tissue-specific expression patterns of *RcTPS* genes. *RcTPS*17, *RcTPS*29 and *RcTPS*45 were not expressed in any of the nine analyzed tissues. Sixteen percent (8 of 49), 6% (3 of 49) and 18% (9 of 49) of the *RcTPS* genes showed the highest transcript abundance in roots, stems and leaves, respectively. Notably, 53% (26 of 49) of the *RcTPS* genes showed the highest transcript abundance in floral organs. Among them, *RcTPS23* showed the highest transcript abundance in the stamen; *RcTPS02*, *RcTPS08*, *RcTPS11*, *RcTPS13*, *RcTPS22*, *RcTPS27*, *RcTPS31*, *RcTPS33*, *RcTPS40* and *RcTPS41* in the pistil; *RcTPS12* in the green petals of the flower bud; *RcTPS20* and *RcTPS37* in the color-changing petals of the flower bud; *RcTPS14* and *RcTPS19* in the pink petals of the flower bud; and *RcTPS01*, *RcTPS06*, *RcTPS09*, *RcTPS16*, *RcTPS26*, *RcTPS34*, *RcTPS35*, *RcTPS36*, *RcTPS44* and *RcTPS46* in the pink petals of the open flower (Figure 7). These data show that the rose *RcTPS* genes are expressed in different organs and in petals at different developmental stages, suggesting their potential roles in the biosynthesis of terpenoids in different organs and during petal development.

### 3.7. Expression Patterns of RcTPS Genes under Abiotic Stress

*RcTPS* genes harbored multiple *cis*-acting elements associated with the stress response. To further address whether the *RcTPS* genes could be regulated by abiotic stresses, we investigated their expression in the open flowers of *R. chinensis* under mannitol-caused osmotic stress (to mimic drought stress ) [65,66,67] and heat treatment at 35 °C. qRT–PCR analyses showed that 18% of the *RcTPS* genes (9 of 49) exhibited a decreasing expression trend under mannitol-induced osmotic stress, while 82% of the *RcTPS* genes (40 of 49) were upregulated upon mannitol treatment (Figure 8a).

After heat treatment at 35 °C for 6 h, 78% of the *RcTPS* (38 of 49) genes were upregulated, but the expression levels of these genes decreased with increasing heat treatment time (Figure 8b). In contrast, 22% of the *RcTPS* genes (11 of 49) exhibited a decreasing trend under heat treatment. These results show that the expression of *RcTPS* genes is regulated by heat and osmotic stresses, which indicates that in roses, the biosynthesis of terpenoids may be affected by abiotic stresses.

## 4. Discussion

Roses are highly appreciated ornamental plants, mainly for their flowers and delicate fragrances [33]. For many years, roses have been used in landscape greening, food, medicine, cosmetic production and especially as cut flowers [68,69,70]. However, roses often suffer from adverse unfavorable environmental conditions such as heat, salinity and drought. Plants produce specialized metabolites to adapt to their environments [71]. Volatile terpenoids play significant roles in resisting abiotic stress conditions [1,4,72]. In previous studies, phytoalexin terpenoids have been reported to accumulate in response to drought, salinity, ultraviolet and elevated CO2 stresses [73,74,75]. Additionally, floral terpenes were reported to protect reproductive organs from oxidative stress [76,77]. Terpene synthases play a leading role in catalyzing the formation of monoterpenes, sesquiterpenes, or diterpenes from the substrates GPP, FPP, or GGPP [3]. Previous studies on floral scent have mainly focused on the characterization and functional verification of the *TPS* genes [78,79,80]. We show here that the expression of *TPS* genes is modified in response to osmotic and heat stresses. 

As a mid-sized gene family, the number of genes in the TPS family in different plant species range from approximately 20 to 150 [3]. In this study, 49 *TPS* genes were identified in the *R*. *chinensis* genome based on BLASTP searches and two conserved TPS domains, and they were divided into five subfamilies (TPS-a, TPS-b, TPS-c, TPS-g and TPS-e/f) based on gene phylogenetic analyses. No members of the TPS-d and TPS-h subfamilies were found in this rose genome. These results are consistent with previous studies which showed that the TPS-d subfamily was specific to gymnosperms and that the TPS-h subfamily was only found in *S. moellendorffii* [3,15,18]. TPS-a is the largest subfamily among RcTPS proteins (34 of 49), followed by TPS-b (6 of 49), which is consistent with previously reported data in *A. thaliana* (22 of the 32 *TPS* genes were *TPS-a* genes), *S. lycopersicum* (15 of the 34 *TPS* genes were *TPS-a* genes), *S. italica* (21 of the 32 *TPS* genes were *TPS-a* genes) and *C.*
*sinensis* (28 of the 55 *TPS* genes were *TPS-a* genes) [18,21,24,25]. Previously, the TPS-a subfamily was reported to be an angiosperm-specific clade, further divided into dicot-specific and monocot-specific subgroups, and that the TPS-a proteins are sesquiterpene synthases or diterpene synthases, while the TPS-b subfamily numbers are responsible for monoterpene or isoprene synthesis [3]. In *A. thaliana*, four TPS-a proteins are cytosolic sesquiterpene synthases, one is a mitochondrial sesquiterpene synthase, and the remaining TPS-a proteins are plastidial or mitochondrial diterpene synthases. Six *A. thaliana* TPS-b proteins are monoterpene synthases that produce myrcene or ocimene [18]. In *S. lycopersicum*, all 15 TPS-a proteins have been reported to be sesquiterpene synthases [20]. In our study, 34 TPS proteins that all belong to the TPS-a subfamily in *R*. *chinensis* were predicted to be sesquiterpene synthases, 10 RcTPS proteins were predicted to be monoterpene synthases, and 5 RcTPS proteins were predicted to be diterpene synthases by the Terzyme database [81]. The 10 putative monoterpene synthases contained 6 TPS-b and 4 TPS-g proteins, and the 5 putative diterpene synthases harbored 1 TPS-c and 4 TPS-e/f proteins. These findings indicate that TPS proteins within the same subfamily may share similar functions. However, further functional analyses are required to confirm this hypothesis.

It was previously reported that plant *TPS* genes generally exhibit conserved intron–exon structures [18,19,82]. In *R*. *chinensis*, we found that 39 out of the 44 *RcTPS* genes of the subfamilies TPS-a, TPS-b and TPS-g contained 7 exons. Five of the remaining *RcTPS* genes contained fewer exons, probably because of intron loss. In contrast, *RcTPS* genes in the subfamilies TPS-c and TPS-e/f contained more exons (between 10 and 14), likely because of the presence of an additional exon encoding the ancestral 200 amino acids of unknown function at the N-terminus [7,18,83]. Previously, the features of RcTPS proteins were found to be related to two domains, one each at the N-terminus (PF01397) and C-terminus (PF03936) [1,84]. All RcTPS-a and RcTPS-b proteins contained the RRX_8_W motif at the N-terminus, a motif shared by monoterpene synthetases in angiosperms that is known to play an important role in monoterpene cyclization [3,17]. None of the RcTPS-g members contained the RRX_8_W motif, since the prominent feature of this subfamily of proteins is the rifeness of acyclic monoterpenes [3]. In addition, it was previously reported that TPS-g proteins can catalyze the biosynthesis of sesquiterpenes and diterpenes [85,86]. The in vivo terpene synthase products mainly depend on their subcellular location and substrate availability [87]. The DDXXD motif was found in all RcTPS proteins. This motif plays an important role in cleaving prenyl diphosphate substrates by binding the metal cofactor (Mg^2+^ or Mn^2+^) at the C-terminus. In addition to the conserved RRX_8_W and DDXXD motifs, NSE/DTE was another important conserved motif at the C-terminus, whose function is the same as that of DDXXD [1,14,15]. NSE/DTE could be found in almost all RcTPS proteins of the subfamilies TPS-a, TPS-b, TPS-g and TPS-e/f.

Expression analysis of *RcTPS* genes in different rose tissues indicated that *RcTPS* genes were mainly expressed in floral organs. *RcTPS*17, *RcTPS*29 and *RcTPS*45 were not expressed in any of the nine analyzed tissues at the analyzed stages, suggesting strict regulation of certain terpenoid biosynthesis genes in roses. Among the 26 *RcTPS* genes with the highest transcript abundance in floral organs (Figure 7), 14 (*RcTPS01, 06*, *08*, *11*, *12*, *14*, *16*, *22*, *23*, *26*, *27*, *34*, *36*, *44*) were predicted to encode putative sesquiterpene synthases, 10 (*RcTPS09, 13*, *19*, *20*, *33*, *35*, *37*, *40*, *41*, *46*) encode putative monoterpene synthases, and 2 (*RcTPS02* and *RcTPS31*) encode putative diterpene synthases. It has been reported in *R*. *chinensis* that the contents of terpenoids are very low in flower buds. When the flowers open, the contents of terpenoids gradually increase and reach a maximum in open flowers, which is consistent with the trend of the transcript abundance of *RcTPS* genes during flowering [88]. These data suggest that *RcTPS* genes play essential roles in the biosynthesis of terpenes in the flower opening process of *R*. *chinensis*. However, previous studies reported that in roses, *NUDX1* is involved in acyclic monoterpene biosynthesis [34,89]. Although, NUDX and TPS pathways relay on a common precursor, namely GPP, the relationship between these two monoterpene biosynthesis pathways remains unclear.

Increasing studies have shown that *TPS* genes are involved in plant development and responses to biotic and abiotic stresses [1,4,72,90]. It has been reported in many plant species that *TPS* genes could be regulated by multiple abiotic stresses and that they may have important roles in resistance to abiotic stresses [24,25,26,27]. Bioinformatics analysis identified multiple *cis*-acting elements that respond to abiotic stresses in the promoter region of *RcTPS* genes, and the average number of *cis*-acting elements related to abiotic stresses in the promoter region of *RcTPS* genes was eight, implying that they might have potential functions in response to abiotic stress. To verify this hypothesis, the expression patterns of *RcTPS* genes under two abiotic stresses were analyzed. Because the water deficit and the associated osmotic stress as well as high-temperature severely limit the growth and flowering of *R*. *chinensis*, we evaluated the expression levels of *RcTPS* genes under osmotic stress induced by mannitol and heat stress by exposure of plants to 35 °C. Our data showed that *RcTPS* genes exhibited different expression levels when the plants were subjected to heat or osmotic stress conditions. Under mannitol-induced osmotic stress and heat stress, the expression levels of most *RcTPS* genes were upregulated, indicating that these genes might be involved in osmotic and heat responses. Among the seven *RcTPS* genes with the highest transcript abundance in fully developed petals in open flowers, *RcTPS06*, *RcTPS16*, *RcTPS34*, *RcTPS*36 and *RcTPS44* were downregulated and *RcTPS46* and *RcTPS01* were upregulated following osmotic and heat stress (Figure 9). Compared to osmotic stress, the peak expression of the *RcTPS* genes under heat treatment occurred relatively early, suggesting that the responses of *RcTPS* genes vary depending on the nature of the stress applied. These data pave the way for more studies to more precisely address the link between the control of terpenoid biosynthesis and the response to abiotic stresses in roses.

## 5. Conclusions

Herein, 49 *TPS* genes were identified in *R*. *chinensis,* a rose species that has played an important role in rose domestication. We divided theses *TPS* genes into five subfamilies. Their chromosomal localization, phylogenetic relationship, gene structure and conserved motifs were subsequently assessed. *Cis*-acting element analysis and expression patterns of *RcTPS* genes under osmotic and heat stresses indicated that *RcTPS* genes were regulated by abiotic stresses and that they might be involved in stress responses in roses. This study provides a foundation for further investigation on the functional role and regulation modes of *RcTPS* genes in roses.

## Figures and Tables

**Figure 1 genes-13-00547-f001:**
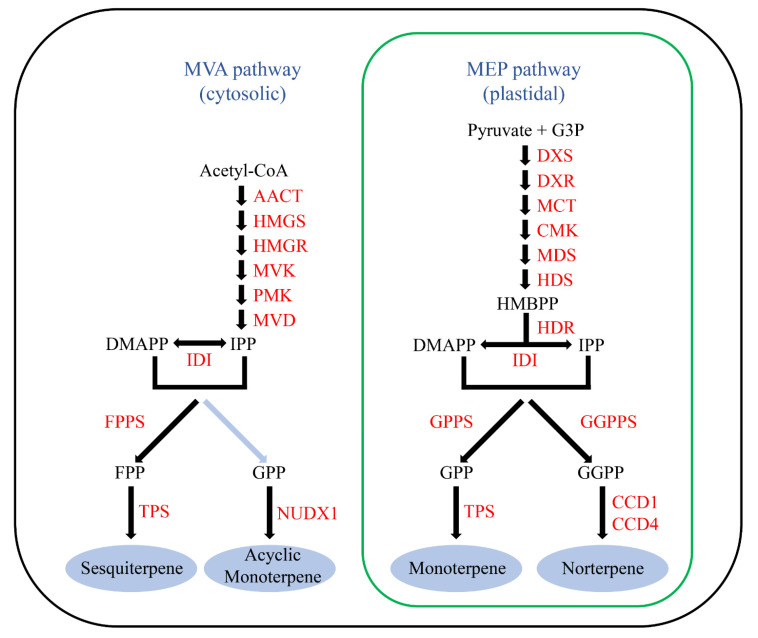
Terpene biosynthesis pathway in the rose. Black arrows indicate that the biosynthetic step has been identified in rose or other species. Blue arrows indicate putative steps with unknown enzymes. AACT, acetoacetyl-CoA thiolase; CCD, carotenoid cleavage dioxygenase; CMK, 4-(cytidine 5’-diphospho)-2-C-methyl-D-erythritol kinase; DMAPP, dimethylallyl diphosphate; DXR, 1-deoxy-D-xylulose 5-phosphate reductoisomerase; DXS, 1-deoxy-D-xylulose 5-phosphate synthase; FPP, farnesyl diphosphate; FPPS, farnesyl diphosphate synthase; G3P, glyceraldehyde 3-phosphate; GGPP, geranylgeranyl diphosphate; GGPPS, geranylgeranyl diphosphate synthase; GPP, geranyl diphosphate; GPPS, geranyl diphosphate synthase; HDR, 1-hydroxy-2-methyl-2-butenyl 4-diphosphate reductase; HDS, 1-hydroxy-2-methyl-2-butenyl 4-diphosphate synthase; HMBPP, 1-hydroxy-2-methyl-2-(E)-butenyl 4-diphosphate; HMGR, 3-hydroxy-3-methylglutaryl CoA reductase; HMGS, 3-hydroxy-3-methylglutaryl CoA synthase; IDI, isopentenyl diphosphate isomerase; IPP, isopentenyl diphosphate; MCT, 2-C-methyl-D-erythritol 4-phosphate cytidyltransferase; MDS, 2-C-methyl-D-erythritol 2,4-cyclodiphosphate synthase; MVD, 5-diphosphomevalonate decarboxylase; MEP, 2-C-methyl-D-erythritol 4-phosphate; MVA, mevalonic acid; MVK, mevalonate kinase; NUDX1, nudix hydrolase1; PMK, 5-phosphomevalonate kinase; TPS, terpene synthase.

**Figure 2 genes-13-00547-f002:**
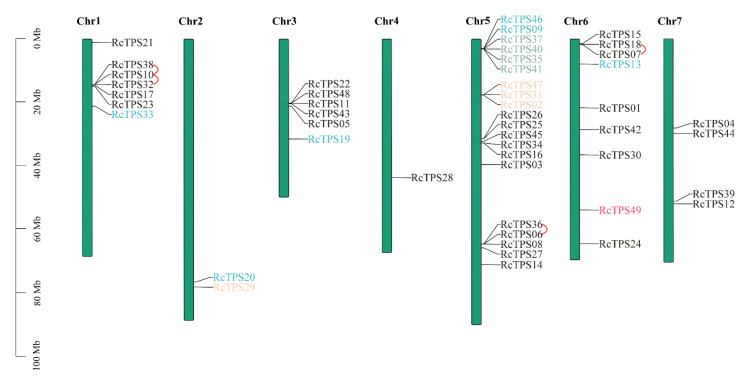
Localization of the 49 TPS genes in R. chinensis 7 chromosomes. The scale bar on the left indicates the size of the chromosome. Gene name colors indicate different subgroups, and red lines indicate tandem repeats.

**Figure 3 genes-13-00547-f003:**
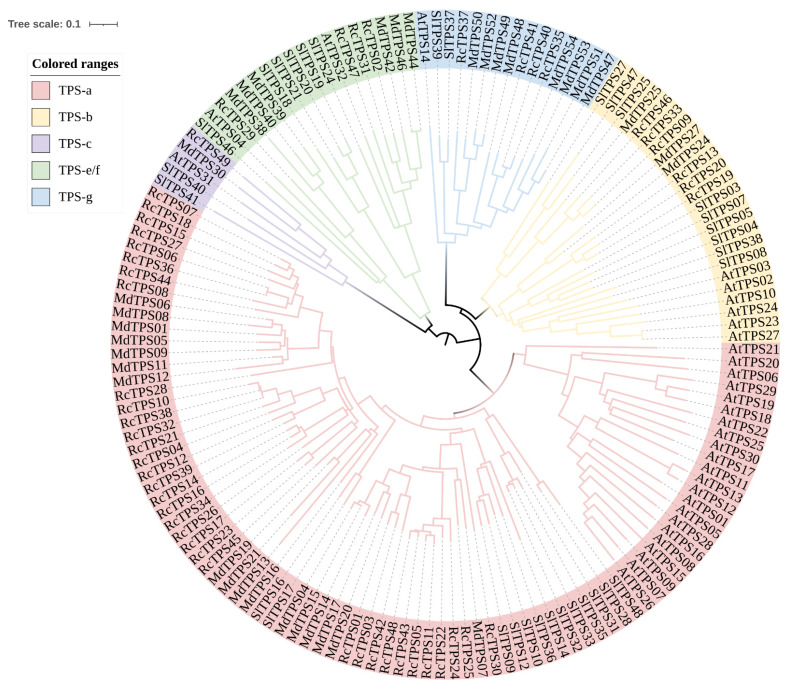
Phylogenetic analysis of TPSs in *R. chinensis*, *A. thaliana* and other species. Subfamilies are highlighted with different colors. Species: *R. chinensis* (Rc); *A. thaliana* (At); *S. lycopersicum* (Sl); *M. domestica* (Md).

**Figure 4 genes-13-00547-f004:**
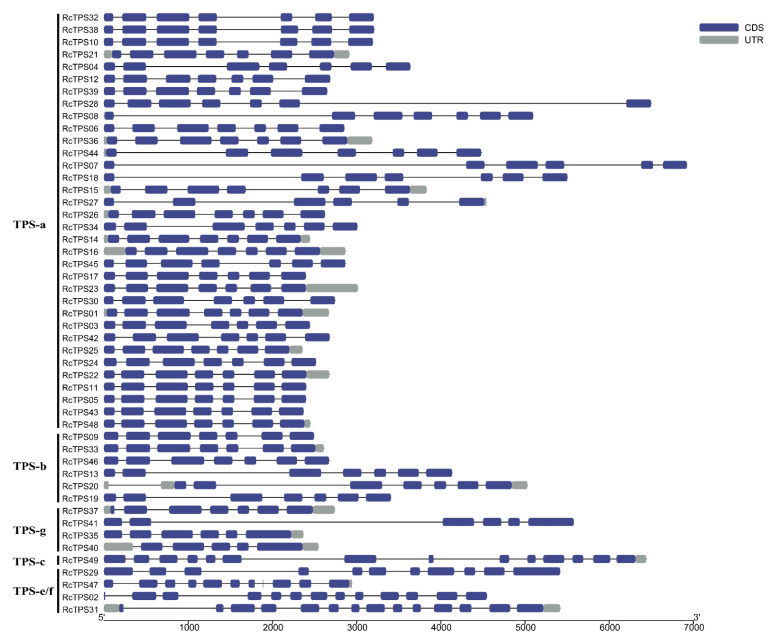
Exon–intron structure analyses of RcTPS genes. The blue and grey boxes indicate CDS and UTR, respectively. Lines indicate introns.

**Figure 5 genes-13-00547-f005:**
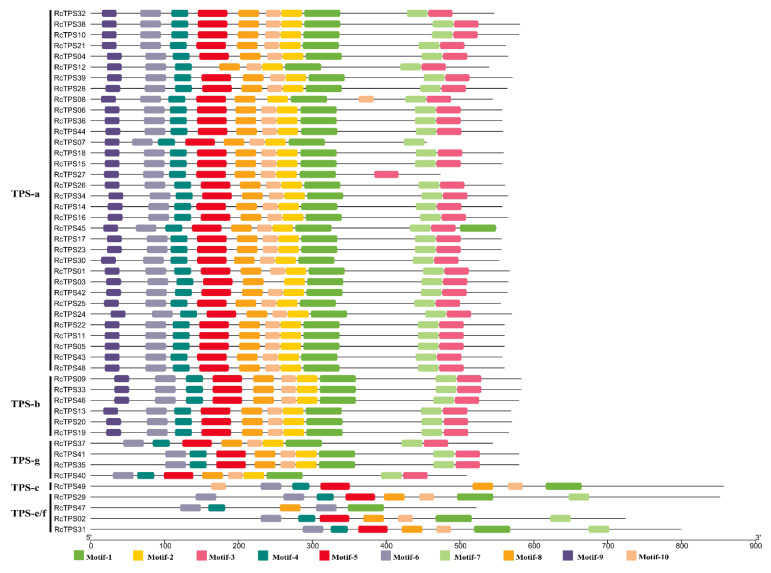
Conserved motifs analyses of RcTPS proteins identified by the MEME program. Motifs are indicted with different color boxes.

**Figure 6 genes-13-00547-f006:**
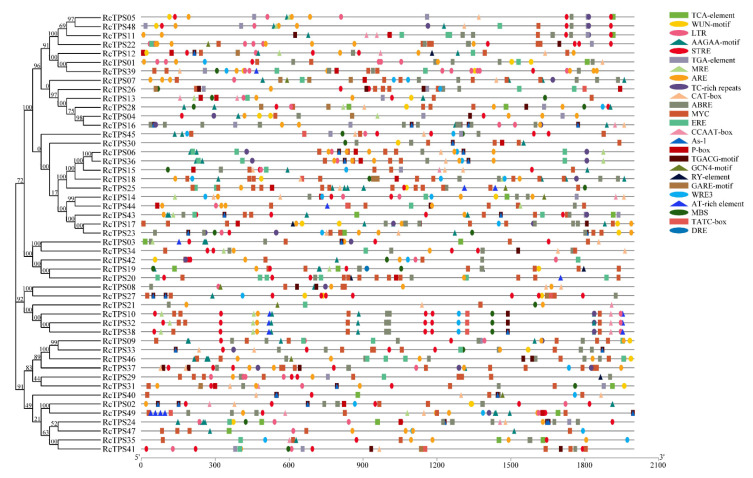
*Cis*-acting elements distribution in the promoter region of *RcTPS* genes. Different color boxes represent different *cis*-acting elements. *Cis*-acting elements related to plant growth and development, stress response and phytohormone response are shown by triangular, oval or rectangular boxes, respectively.

**Figure 7 genes-13-00547-f007:**
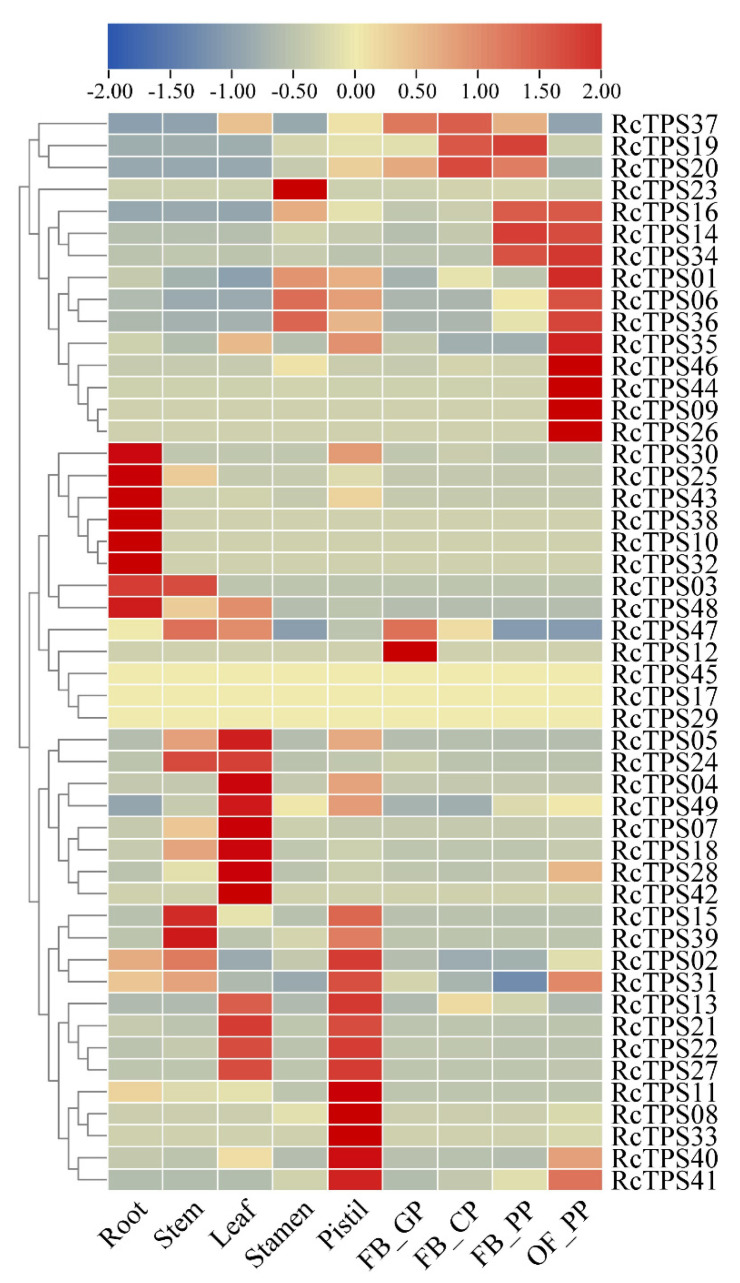
Heatmap of the expression profiles of RcTPS genes in different R. chinensis organs generated using TBtools software. The expression abundance of each transcript is represented by the color bar: red, higher expression; blue, lower expression. FB_GP: green petals in the flower buds; FB_CP: color-changing petals in the flower buds; FB_PP: pink petals in the flower buds; OF_PP: pink petals of the open flowers.

**Figure 8 genes-13-00547-f008:**
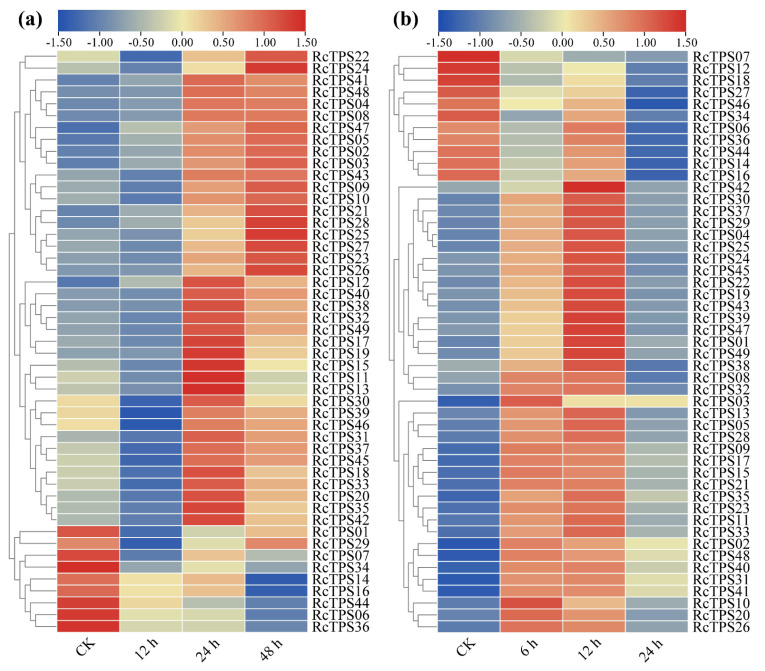
Expression profiles of RcTPS genes in petals of fully open flowers of R. chinensis under osmotic and heat stresses. (**a**) Expression profiles of RcTPS genes in response to 200 mM mannitol treatment for 48 h. (**b**) Expression profiles of RcTPS genes in response to heat treatment (35 °C) for 24 h. Heatmap was drawn using TBtools software.

**Figure 9 genes-13-00547-f009:**
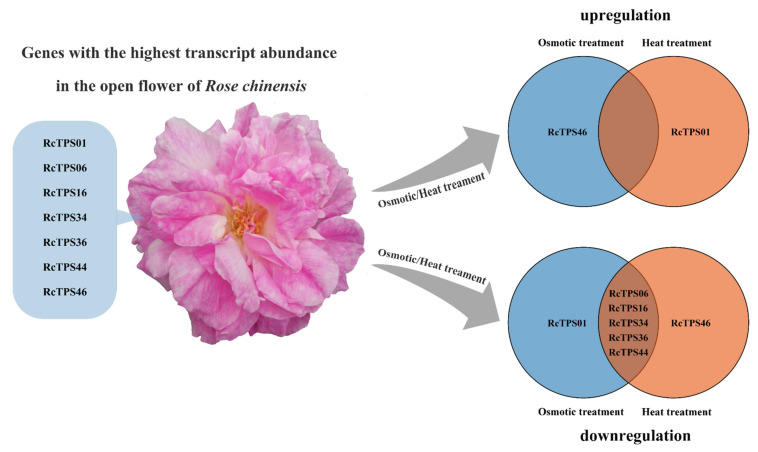
Expression patterns of the *RcTPS* genes with the highest transcript abundance in petals of fully open flowers under osmotic (200 mM mannitol) and heat (35 °C) treatment.

**Table 1 genes-13-00547-t001:** Characterization of TPS family identified in *R. chinensis*.

Gene	Accession Number	Chr	AA (aa)	MW (kDa)	pI	Subcellular Localization	Subfamily
*RcTPS01*	RchiOBHm_Chr6g0265741	6	567	65.92	5.16	Chloroplast ^a^/Cytoplasm ^b^	TPS-a
*RcTPS02*	RchiOBHm_Chr5g0023641	5	724	81.88	5.72	Chloroplast ^a, b^	TPS-e/f
*RcTPS03*	RchiOBHm_Chr5g0044191	5	565	65.40	5.25	Chloroplast ^a^/Cytoplasm ^b^	TPS-a
*RcTPS04*	RchiOBHm_Chr7g0210371	7	565	65.34	5.07	Chloroplast ^a^/Cytoplasm ^b^	TPS-a
*RcTPS05*	RchiOBHm_Chr3g0475221	3	560	64.87	5.17	Chloroplast ^a, b^	TPS-a
*RcTPS06*	RchiOBHm_Chr5g0059511	5	557	64.72	5.95	Chloroplast ^a^/Cytoplasm ^b^	TPS-a
*RcTPS07*	RchiOBHm_Chr6g0246011	6	455	52.89	6.07	Chloroplast ^a^/Cytoplasm ^b^	TPS-a
*RcTPS08*	RchiOBHm_Chr5g0059541	5	544	62.75	5.19	Chloroplast ^a^/Cytoplasm ^b^	TPS-a
*RcTPS09*	RchiOBHm_Chr5g0004631	5	583	67.00	5.14	Chloroplast ^a, b^	TPS-b
*RcTPS10*	RchiOBHm_Chr1g0326061	1	580	66.90	5.22	Chloroplast ^a^/Cytoplasm ^b^	TPS-a
*RcTPS11*	RchiOBHm_Chr3g0474501	3	560	64.58	5.17	Chloroplast ^a, b^	TPS-a
*RcTPS12*	RchiOBHm_Chr7g0228501	7	539	62.16	5.43	Chloroplast ^a^/Cytoplasm ^b^	TPS-a
*RcTPS13*	RchiOBHm_Chr6g0252721	6	569	65.74	5.59	Chloroplast ^a^/Cytoplasm ^b^	TPS-b
*RcTPS14*	RchiOBHm_Chr5g0065101	5	557	64.23	5.18	Chloroplast ^a^/Cytoplasm ^b^	TPS-a
*RcTPS15*	RchiOBHm_Chr6g0245751	6	557	64.36	5.35	Chloroplast ^a^/Cytoplasm ^b^	TPS-a
*RcTPS16*	RchiOBHm_Chr5g0038101	5	565	65.25	5.66	Chloroplast ^a^/Cytoplasm ^b^	TPS-a
*RcTPS17*	RchiOBHm_Chr1g0326251	1	556	64.54	5.21	Chloroplast ^a^/Cytoplasm ^b^	TPS-a
*RcTPS18*	RchiOBHm_Chr6g0246001	6	559	64.70	5.57	Chloroplast ^a^/Cytoplasm ^b^	TPS-a
*RcTPS19*	RchiOBHm_Chr3g0484891	3	566	64.99	5.49	Chloroplast ^a^/Cytoplasm ^b^	TPS-b
*RcTPS20*	RchiOBHm_Chr2g0160561	2	570	65.43	5.62	Chloroplast ^a^/Cytoplasm ^b^	TPS-b
*RcTPS21*	RchiOBHm_Chr1g0313881	1	562	65.05	5.11	Chloroplast ^a^/Cytoplasm ^b^	TPS-a
*RcTPS22*	RchiOBHm_Chr3g0474411	3	560	64.74	5.15	Chloroplast ^a^/Cytoplasm ^b^	TPS-a
*RcTPS23*	RchiOBHm_Chr1g0326391	1	556	64.54	5.37	Chloroplast ^a^/Cytoplasm ^b^	TPS-a
*RcTPS24*	RchiOBHm_Chr6g0305391	6	570	65.88	5.07	Chloroplast ^a, b^	TPS-a
*RcTPS25*	RchiOBHm_Chr5g0037011	5	555	64.33	5.25	Chloroplast ^a, b^	TPS-a
*RcTPS26*	RchiOBHm_Chr5g0036921	5	561	65.14	5.36	Chloroplast ^a^/Cytoplasm ^b^	TPS-a
*RcTPS27*	RchiOBHm_Chr5g0060571	5	473	54.87	5.86	Chloroplast ^a^/Cytoplasm ^b^	TPS-a
*RcTPS28*	RchiOBHm_Chr4g0418071	4	564	64.74	5.16	Chloroplast ^a^/Cytoplasm ^b^	TPS-a
*RcTPS29*	RchiOBHm_Chr2g0162311	2	852	96.91	6.12	Chloroplast ^a, b^	TPS-e/f
*RcTPS30*	RchiOBHm_Chr6g0274871	6	553	63.26	5.49	Chloroplast ^a^/Cytoplasm ^b^	TPS-a
*RcTPS31*	RchiOBHm_Chr5g0023471	5	799	90.14	5.58	Chloroplast ^a, b^	TPS-e/f
*RcTPS32*	RchiOBHm_Chr1g0326071	1	546	62.85	5.37	Chloroplast ^a^/Cytoplasm ^b^	TPS-a
*RcTPS33*	RchiOBHm_Chr1g0331211	1	583	67.49	5.28	Chloroplast ^a, b^	TPS-b
*RcTPS34*	RchiOBHm_Chr5g0038021	5	565	65.52	5.63	Chloroplast ^a^/Cytoplasm ^b^	TPS-a
*RcTPS35*	RchiOBHm_Chr5g0004761	5	580	66.13	6.17	Chloroplast ^a^/Cytoplasm ^b^	TPS-g
*RcTPS36*	RchiOBHm_Chr5g0059501	5	557	64.72	5.95	Chloroplast ^a^/Cytoplasm ^b^	TPS-a
*RcTPS37*	RchiOBHm_Chr5g0004711	5	544	62.64	5.09	Chloroplast ^a, b^	TPS-g
*RcTPS38*	RchiOBHm_Chr1g0326051	1	581	67.11	5.34	Chloroplast ^a^/Cytoplasm ^b^	TPS-a
*RcTPS39*	RchiOBHm_Chr7g0227831	7	571	66.02	5.26	Chloroplast ^a^/Cytoplasm ^b^	TPS-a
*RcTPS40*	RchiOBHm_Chr5g0004731	5	509	58.29	5.26	Chloroplast ^a^/Cytoplasm ^b^	TPS-g
*RcTPS41*	RchiOBHm_Chr5g0004801	5	580	66.55	6.05	Chloroplast ^a^/Cytoplasm ^b^	TPS-g
*RcTPS42*	RchiOBHm_Chr6g0270581	6	564	65.34	5.5	Chloroplast ^a^/Cytoplasm ^b^	TPS-a
*RcTPS43*	RchiOBHm_Chr3g0474541	3	557	64.31	5.06	Chloroplast ^a, b^	TPS-a
*RcTPS44*	RchiOBHm_Chr7g0212441	7	558	64.77	5.46	Chloroplast ^a^/Cytoplasm ^b^	TPS-a
*RcTPS45*	RchiOBHm_Chr5g0037601	5	549	63.56	5.23	Chloroplast ^a^/Cytoplasm ^b^	TPS-a
*RcTPS46*	RchiOBHm_Chr5g0004591	5	580	66.74	5.16	Chloroplast ^a, b^	TPS-b
*RcTPS47*	RchiOBHm_Chr5g0023441	5	522	59.69	8.91	Chloroplast ^a, b^	TPS-e/f
*RcTPS48*	RchiOBHm_Chr3g0474441	3	560	64.75	5.08	Chloroplast ^a^/Cytoplasm ^b^	TPS-a
*RcTPS49*	RchiOBHm_Chr6g0290871	6	857	98.52	5.76	Chloroplast ^a, b^	TPS-c

Accession Number, it is annotated in *R. chinensis* genome (https://lipmbrowsers.toulouse.inra.fr//pub/RchiOBHm-V2/, accessed on 24 February 2022) [31]; Chr, Chromosome; AA, amino acids; MW, molecular weight; pI, theoretical isoelectric point; Subcellular localization is predicted by Plant-mPLoc (http://www.csbio.sjtu.edu.cn/bioinf/plantmulti/, accessed on 24 February 2022) and pLoc-mPlant (http://www.jcibioinfo.cn/pLoc-mPlant/, accessed on 24 February 2022) tools. ^a, b^ indicates the result of Plant-mPLoc and pLoc-mPlant, respectively.

## Data Availability

Accession numbers of all the *TPS* genes used in this study and the candidate *RcTPS* genes are listed in Appendix A and Table 1, respectively. The rose genome database is available at: https://lipmbrowsers.toulouse.inra.fr//pub/RchiOBHm-V2/, accessed on 24 February 2022.

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
