# Peer review of "Tissue-Specific Expression of the Terpene Synthase Family Genes in Rosa chinensis and Effect of Abiotic Stress Conditions"

_genes, 2022, doi:10.3390/genes13030547_

Round 1

Reviewer 1 Report

Article readable in English, abundance of methods used for data analysis, excellent and illustrative figures relating to the results, could add some figures on the pathways described or on Rosa chinensis in the introduction.

Results consistent with what they wanted to demonstrate.

They could better specify the practical benefits of the study.

Line 399-401 because they chose fully open flowers to analyze the data rather than half open as stated in the discussion.

Author Response

Response to Reviewer 1 Comments

Point 1: Article readable in English, abundance of methods used for data analysis, excellent and illustrative figures relating to the results, could add some figures on the pathways described or on Rosa chinensis in the introduction。

Response 1: Thank you for your review and endorsement of this article. According to your suggestion, we have added one figure on the pathways described or on Rosa chinensis (lines 99-112) and the related description (lines 74-78)

Point 2: Results consistent with what they wanted to demonstrate.

They could better specify the practical benefits of the study.

Response 2: Thanks for your thoughtful comments. We completed the introduction section about the biological functions of TPS and how TPS participate in response to stress conditions (lines 83-98).

Point 3: Line 399-401 because they chose fully open flowers to analyze the data rather than half open as stated in the discussion

Response 3: Thanks for your careful review, we have made the corresponding modification (lines 431).

Reviewer 2 Report

In my opinion Authors present important information related to terpene synthases phylogeny, expression, gene organization and response to heat and drought stress in Rosa chinensis. Study is well planned and performed. Results of research support well obtained conclusions.

Introduction

Authors could add to the Introduction section a short information of sample terpene compounds that are present in Rosa chinensis- their putative role for example in response to abiotic or biotic stress conditions, function as volatile fragrances to attract pollinating insect as bee or other important biological  functions.

Authors could shortly describe how (putative mechanism) terpene synthases participate in response to stress conditions.

Section 2.7

More detailed description of RT-PCR experiments is necessary:

How the quality of RNA was assessed?

Amount of RNA that was taken for analysis.

Conditions of reverse transcription reaction.

The citation of previous use of reference gene or analysis of its expression stability using software as bestKeeper or related.

The remnants of genomic DNA should be removed by DNaseI digestion- provide conditions of such step.

Citation of Livak and Schmittgen  method used for relative gene expression calculation should be included.

Discussion

Authors correctly performed searches for cis-active elements within the promoters- 2kb upstream from the translation initiation methionine. However, most of the in vivo functional cis-active elements (80%) is concentrated within proximal promoters- usually several hundreds of bp from transcription start site. Authors could add one or two sentences to show this fact, sample reference are below:

Keilwagen J, Grau J, Paponov IA, Posch S, Strickert M, Grosse I (2011) De-novo discovery of differentially abundant transcription factor binding sites including their positional preference. PLoS Comput Biol 7:e1001070. doi: 10.1371/journal.pcbi.1001070.

Yu ChP, Lin JJ, Li WH (2016) Positional distribution of transcription factor binding sites in Arabidopsis thaliana. Sci Rep 6:25164.

Molina, C.; Grotewold, E. Genome wide analysis of Arabidopsis core promoters. BMC Genomics 2005, 6, 25. doi: 10.1186/1471-2164-6-25

Other

Citations in lines 350-351 should be as numbers  inside “[]”.

Author Response

Response to Reviewer 2 Comments

Point 1: Authors could add to the Introduction section a short information of sample terpene compounds that are present in Rosa chinensis- their putative role for example in response to abiotic or biotic stress conditions, function as volatile fragrances to attract pollinating insect as bee or other important biological functions.

Authors could shortly describe how (putative mechanism) terpene synthases participate in response to stress conditions.。

Response 1: Thank you for your review and endorsement of this article. According to your suggestion, We completed the introduction section about the biological functions of TPS and how TPS participate in response to stress conditions (lines 83-89).

Point 2: More detailed description of RT-PCR experiments is necessary:

How the quality of RNA was assessed?

Amount of RNA that was taken for analysis.

Conditions of reverse transcription reaction.

The citation of previous use of reference gene or analysis of its expression stability using software as bestKeeper or related.

The remnants of genomic DNA should be removed by DNaseI digestion- provide conditions of such step.

Citation of Livak and Schmittgen method used for relative gene expression calculation should be included.

Response 2: Thanks for your thoughtful comments. we have added the related descriptions about RNA extraction (lines 125-130). Reference about housekeeping gene and method used for relative gene expression calculation used in this study have been added (lines 194 and 196).

Lu, J.; Sun, J.; Jiang, A.; Bai, M.; Fan, C.; Liu, J.; Ning, G.; Wang, C. Alternate expression of CONSTANS-LIKE 4 in short days and CONSTANS in long days facilitates day-neutral response in Rosa chinensis. Journal of Experimental Botany 2020, 71, 4057-4068, doi:10.1093/jxb/eraa161.

Zhang, X.; Wang, Q.; Yang, S.; Lin, S.; Bao, M.; Bendahmane, M.; Wu, Q.; Wang, C.; Fu, X. Identification and Characterization of the MADS-Box Genes and Their Contribution to Flower Organ in Carnation (Dianthus caryophyllus L.). Genes 2018, 9, doi:10.3390/genes9040193.

Point 3: Authors correctly performed searches for cis-active elements within the promoters- 2kb upstream from the translation initiation methionine. However, most of the in vivo functional cis-active elements (80%) is concentrated within proximal promoters- usually several hundreds of bp from transcription start site. Authors could add one or two sentences to show this fact, sample reference are below:

Keilwagen J, Grau J, Paponov IA, Posch S, Strickert M, Grosse I (2011) De-novo discovery of differentially abundant transcription factor binding sites including their positional preference. PLoS Comput Biol 7:e1001070. doi: 10.1371/journal.pcbi.1001070.

Yu ChP, Lin JJ, Li WH (2016) Positional distribution of transcription factor binding sites in Arabidopsis thaliana. Sci Rep 6:25164.

Molina, C.; Grotewold, E. Genome wide analysis of Arabidopsis core promoters. BMC Genomics 2005, 6, 25. doi: 10.1186/1471-2164-6-25

Response 3: Thanks for your thoughtful comments, we have added the corresponding description (lines 289-291).

Point 4: Citations in lines 350-351 should be as numbers  inside “[]”.

Response 4: Thanks for your careful review. We have made the correction (lines 380-381).
